# Aquorin Bioluminescence-Based Ca^2+^ Imaging Reveals Differential Calcium Signaling Responses to Abiotic Stresses in *Physcomitrella patens*

**DOI:** 10.3390/plants14081178

**Published:** 2025-04-10

**Authors:** Jiamin Shen, Kexin Ding, Zhiming Yu, Yuzhen Zhang, Jun Ni, Yuhuan Wu

**Affiliations:** 1College of Life and Environmental Sciences, Hangzhou Normal University, Hangzhou 311121, China; 15968130773@163.com (J.S.); dkx020966@163.com (K.D.); yuzhiming@hznu.edu.cn (Z.Y.); 2022111010074@stu.hznu.edu.cn (Y.Z.); 2Zhejiang Provincial Key Laboratory for Genetic Improvement and Quality Control of Medicinal Plants, Hangzhou Normal University, Hangzhou 311121, China

**Keywords:** abiotic stresses, cold, drought, inhibitor, ROS, salt

## Abstract

Calcium ions (Ca^2+^) are an important secondary messenger in plant signal transduction networks. The cytosolic free Ca^2+^ concentration ([Ca^2+^]_i_) of plants changes rapidly when they are subjected to different abiotic stresses, which drives calcium signaling. Although this process has been extensively studied in spermatophytes, the details of calcium signaling in bryophytes remains largely unknown. In our study, we reconstituted aequorin in the bryophyte *Physcomitrella patens*, optimized the percentage of ethanol in the Ca^2+^ discharging solution, and measured the [Ca^2+^]_i_ changes induced by different stresses. In addition, we observed that the sources of Ca^2+^ accessed following exposure to cold, drought, salt, and oxidative stress were different. Furthermore, we showed that long-term saline environments could suppress the basal [Ca^2+^]_i_ of *P. patens*, and the peak value of [Ca^2+^]_i_ induced by different stresses was lower than that of plants growing in non-stressed environments. This is the first systematic study of calcium signaling in bryophytes, and we provided an efficient and convenient tool to study calcium signaling in response to different abiotic stresses in bryophytes.

## 1. Introduction

Bryophytes are an important component of biodiversity, with approximately 21,000 species worldwide [1,2]. These small plants often act as ecosystem pioneers, and they play other important roles in enabling the monitoring of environmental pollution, protecting soil and water resources, maintaining material cycles and energy flows, and sequestering carbon [3,4,5,6,7]. The ongoing environmental and climatic changes pose a high risk of extinction for some bryophytes [8]. Thus, the study of how bryophytes respond to environmental stress is of great theoretical and practical importance for the conservation and use of species resources and for the maintenance of ecological balance.

Over the course of their evolution, plants have adapted to environmental changes through a series of molecular mechanisms that couple stimulus signals to the specific physiological effects they induce. Calcium ions (Ca^2+^) are an important secondary messenger in plant signal transduction networks in response to a large number of environmental stimuli such as cold, drought, salt, and oxidative stress [9,10]. Upon exposure to environmental signals, Ca^2+^ enters the cell from calcium pools (both intracellular and extracellular) via Ca^2+^ channels and transporter proteins [11], resulting in a rapid increase in the concentration of cytosolic free Ca^2+^ ([Ca^2+^]_i_). This generates a unique Ca^2+^ signal that interacts with downstream effector proteins, such as calmodulin and calcium-dependent protein kinases, to activate their downstream targets and induce short- and long-term adaptive responses to specific environmental conditions [10,12,13,14]. Changes in [Ca^2+^]_i_ are an early cellular response to specific environmental stimuli, and plants vary in their [Ca^2+^]_i_ changes in response to different external stimuli [15]. Excessive [Ca^2+^]_i_ is toxic to cells, and so plants maintain their [Ca^2+^]_i_ homeostasis by transferring Ca^2+^ to organelles such as vesicles and the endoplasmic reticulum (ER) via Ca^2+^-ATPases and H^+^/Ca^2+^-antiporters [16,17,18].

As the [Ca^2+^]_i_ signal is generated transiently, the effective detection of [Ca^2+^]_i_ in real time is important for studying plant physiology. Genetically encoded calcium indicators (GECIs) are widely used to detect changes in [Ca^2+^]_i_ in plants [19,20]. Aequorin, a calcium-sensitive photoprotein from the jellyfish *Aequoria victoria* [21], is formed from an oxygen molecule; a single polypeptide chain, apoaequorin; and a hydrophobic luminophore, coelenterazine [22,23]. Each apoaequorin binds three Ca^2+^ ions, which causes coelenterazine to dissociate from the complex and emit blue light at a wavelength of 469 nm [24]. The higher the [Ca^2+^]_i_, the stronger the luminescence; therefore, the [Ca^2+^]_i_ can be deduced from the luminescence intensity.

In this paper, we transferred the apoaequorin cDNA into *Physcomitrella patens* and reconstituted aequorin in vitro. For the first time, we employed aequorin-based bioluminescence Ca^2+^ imaging to systematically explore the changes of [Ca^2+^]_i_ in response to diverse environmental stresses in *P. patens*. Our findings provide a foundation for the future comparative studies of calcium signaling evolution between bryophytes and angiosperm models (*Arabidopsis* and rice) through the analysis of conserved and divergent response patterns.

## 2. Results

### 2.1. Production of Transgenic P. patens Expressing Apoaequorin

To examine the responses of [Ca^2+^]_i_ to abiotic stresses, we prepared transgenic *P. patens* expressing apoaequorin under the control of the *PpEF1-α* promoter (Figure 1A). The heterologous expression of apoaequorin had no significant effect on the growth or development of moss (Appendix A). The PCR and RT-PCR results indicated that the *APOAEQUORIN* gene was successfully inserted into specific sites in the genome of *P. patens,* and that it was successfully expressed under the *PpEF1-α* promoter (Figure 1B). All three transformants contained an insertion at the same location, and they did not differ substantially at the genomic level; thus, one transformant, *AQ-OE1*, was chosen for the subsequent calcium signaling assay.

### 2.2. Optimization of the Discharging Solution for Luminescence Imaging in P. patens

The discharging solution treatment causes the release of all intracellular Ca^2+^, which is important for the calculation of [Ca^2+^]_i_ based on luminescence intensity. In this study, *P. patens* was treated with a series of discharging solution with different ethanol concentration gradients. At ethanol levels below 15%, the luminescence intensity increased significantly with increasing ethanol content; when the plants were treated with a discharging solution containing 15% ethanol, the plants emitted approximately 1.5 times more luminescence than when the discharging solution with 0% ethanol was used. The luminescence intensity was significantly lower in plants treated with a discharging solution containing more than 15% ethanol, although the intensity did not change significantly as the ethanol content increased above this threshold (Figure 2). A discharging solution with an ethanol content of 15% was thus used as the optimum treatment solution for *P. patens*.

### 2.3. Cold Induces an Immediate [Ca^2+^]_i_ Spike in P. patens

The luminescence intensity was related to the degree of cold stress: the lower the temperature, the greater the luminescence intensity, with temperatures of 15 °C and below inducing a strong luminescence signal (Figure 3A). The luminescence signals under the 20 °C treatment and in the control (25 °C) were not significantly different, whereas the luminescence signals induced by 15 °C, 10 °C, 5 °C, and 0 °C were significantly different from those of the control (*p* < 0.001) (Figure 3B). The fluorescence values (*L*/*L_max_*) were consistent with the overall trend of average luminescence intensity, with *L*/*L_max_* increasing as the temperature decreased. A further analysis revealed no significant difference between the *L*/*L_max_* values under the control and 20 °C treatment or the 15 °C treatment, whereas the 10 °C, 5 °C, and 0 °C treatments caused a significant difference relative to the control (*p* < 0.01) (Figure 3B).

The basic parameters of the [Ca^2+^]_i_ response were analyzed, revealing that the calcium signal in *P. patens* appeared quickly after the 5 °C and 0 °C treatment, with a significant increase in luminescence intensity and *L*/*L_max_*, which peaked after around 15 s (seconds). The maximum values of luminescence intensity and *L*/*L_max_* induced by the 0 °C treatment were 3.5 times and 5 times higher than those for the 5 °C treatment, respectively, and decreased rapidly within 30 s, followed by a return to basal levels within 3 min. In contrast, the peaks of luminescence intensity and *L*/*L_max_* induced by the 10 °C, 15 °C, and 20 °C treatments appeared later, at 60–75 s, and the increase was relatively weak compared with the resting state (Figure 3C,D). In general, the greater the degree of cold stress, the faster the appearance of the luminescence signal, and the greater the luminescence intensity and *L*/*L_max_*.

### 2.4. Drought Induces a [Ca^2+^]_i_ Spike in P. patens

The greater the degree of drought stress, the stronger the luminescence signal (Figure 4A). There was no significant difference in the average luminescence intensity induced by treatments with lower concentrations of mannitol (0.15–0.3 M) when compared with the control group; however, the average luminescence intensity under 0.5–1 M mannitol was very different from that of the control group (*p* < 0.001). The luminescence intensity increased slowly in the range of 0–0.3 M (low-concentration) mannitol, whereas the higher mannitol concentrations caused a rapid increase in luminescence intensity (Figure 4B). The fluorescence value (*L*/*L_max_*) was consistent with the trend of average luminescence intensity, with no significant difference between the 0.15–0.4 M treatments and the control. When the concentration increased to 0.5 M, the *L*/*L_max_* increased to significantly more than the control group (*p* < 0.001), and further increases in mannitol rapidly increased the *L*/*L_max_*. In the range of 0.5–0.9 M mannitol, the *L*/*L_max_* values were significantly different for each mannitol concentration (*p* < 0.001), whereas the results of 0.9 M and 1 M treatment were similar (Figure 4B). Higher concentrations of mannitol (0.5–0.9 M) thus induced a large Ca^2+^ influx, rapidly increasing drought-induced [Ca^2+^]_i_, whereas mannitol treatments below this concentration range had little effect on the [Ca^2+^]_i_ of *P. patens*. Mannitol concentrations above 0.9 M did not induce further increases in calcium signaling.

Drought stress triggered rapid [Ca^2+^]_i_ transients in *P. patens*, with response kinetics and amplitude strongly dependent on mannitol concentration. After treatment with the 0.75–1 M mannitol solution, the luminescence intensity and the *L*/*L_max_* increased significantly, peaking after about 15 s. Most of the luminescence signals were collected in the first 90 s, and then slowly returned to the basic state within 3 min. By contrast, lower concentrations of 0.15–0.5 M mannitol induced a slower increase in luminescence, reaching a peak at about 45–60 s (Figure 4C,D).

### 2.5. NaCl Induces a [Ca^2+^]_i_ Spike in P. patens

Similarly to the previous reports in rice [9], the [Ca^2+^]_i_-dependent luminescence signal was related to the degree of salt stress (Figure 5A); 0.1 M NaCl induced a significantly stronger average luminescence signal than the control (*p* < 0.05), and a highly significant difference was observed between the signals induced under the 0.1 M, 0.2 M, and 0.3 M NaCl treatments (*p* < 0.001), indicating that low concentrations of NaCl induced a rapid increase in [Ca^2+^]_i_-dependent luminescence signaling. The luminescence signal induced by 1 M NaCl was greater than those induced by the lower salt concentrations (*p* < 0.001), and 2 M NaCl induced a further significant increase in the luminescence signal compared with 1 M NaCl (*p* < 0.05) (Figure 5B). A detailed analysis of the *L*/*L_max_* results showed no significant difference for any of the 0.1–0.3 M NaCl treatments compared with the control, but a significant increase in *L*/*L_max_* under the 0.4 M NaCl treatment (*p* < 0.05). The *L*/*L_max_* increased significantly with increasing NaCl concentrations, with the 2 M NaCl treatment inducing the greatest *L*/*L_max_* value (*p* < 0.001), which was 3.3 times higher than that under the 1 M NaCl treatment (Figure 5B). This high value under the 2 M NaCl treatment may be caused by the abnormally low luminescence signal detected when these plants were treated with discharging solution. We verified this by treating the plants with 2 M NaCl solution for 30 s. Upon dissecting the leaves to make mounts and observing them under the microscope, we saw no difference in the leaf’s overall morphology. Despite this, there was a high efflux of cell contents and a high degree of coalescence in the material remaining in the cells, indicating that the cells had been damaged by the high NaCl concentration, thereby affecting the treatment results ( Appendix A).

NaCl-induced [Ca^2+^]_i_ transients exhibited concentration-dependent temporal patterns. The luminescence signals induced by the lower concentrations of 0.1–0.3 M NaCl appeared slowly, and the luminescence intensity and *L*/*L_max_* peaked at about 60 s before returning to the resting state at about 120 s. After the 0.4 M NaCl treatment, the luminescence intensity and *L*/*L_max_* reached a maximum at about 45 s, whereas the luminescence signal could not be detected by about 120 s. After treatment with 0.5–1 M NaCl, the luminescence signal peaked at about 15 s, initially decreased rapidly, and then slowly returned to a resting state. After the treatment with 2 M NaCl, the luminescence signal could still be detected after 210 s, indicating that the [Ca^2+^]_i_ could not drop to basal levels and that the intracellular calcium transport system may be disrupted at this NaCl concentration (Figure 5C,D). This result is consistent with the previous results (Appendix A), indicating that the cell structure is damaged by this high NaCl level, which also affects calcium signaling.

### 2.6. H_2_O_2_ Induced a [Ca^2+^]_i_ Spike in P. patens

In this study, we used H_2_O_2_, a reactive oxygen species (ROS), to simulate the oxidation environment by detecting the changes in calcium signaling in response to different H_2_O_2_ concentrations. In general, the luminescence signal increased as the H_2_O_2_ concentration increased (Figure 6A). The 2 mM H_2_O_2_ treatment induced a luminescence signal significantly higher than that in the control group (*p* < 0.01), and 4 mM H_2_O_2_ induced a significantly greater signal than 2 mM H_2_O_2_ (*p* < 0.001) (Figure 6B). The 5 mM, 8 mM, and 10 mM H_2_O_2_ treatments induced luminescence signals that were similar to each other but significantly higher than those seem with the 4 mM H_2_O_2_ treatment (*p* < 0.01). The *L*/*L_max_* of *P. patens* in response to oxidative stress was consistent with the overall trend of average luminescence intensity (Figure 6B).

H_2_O_2_-induced [Ca^2+^]_i_ transients exhibited distinct kinetic profiles compared to other stressors, characterized by delayed onset and prolonged duration. The average luminescence intensity and *L*/*L_max_* revealed that the H_2_O_2_-induced luminescence signal appeared slowly, peaking at about 45 s under the 2–10 mM H_2_O_2_ treatments, and then slowly decreased, returning after 4–5 min to the resting state, which was then maintained for a long time (Figure 6C,D). The dynamics of the H_2_O_2_-induced calcium signals differed from those induced by the other stresses analyzed, with the [Ca^2+^]_i_ peaks appearing later, falling slowly, and taking longer to return to the resting state. In addition, we examined the effect of a 10 mM H_2_O_2_ treatment every 30 s over a longer time period (45 min). The luminescence signal was strongest in the first 2 min, with a weak luminescence signal observed at 3–4 min; no luminescence signal was collected in the subsequent 40 min (Appendix A).

### 2.7. Different Sources of Ca^2+^ Induced by Various Stresses

Pharmacological inhibition experiments revealed distinct Ca^2+^ channel dependencies for different stress-induced [Ca^2+^]_i_ responses. All the inhibitors examined had a significant dose-dependent effect on the cold-induced [Ca^2+^]_i_ increase, and 1 mM GdCl_3_, LaCl_3_, neomycin and thapsigargin almost completely inhibited the [Ca^2+^]_i_ increase (Figure 7A). In contrast to the significant effect of four inhibitors to cold (10 °C)-induced [Ca^2+^]_i_ increase, GdCl_3_, neomycin and thapsigargin treatments had mild but significant effects on the drought (0.5 M mannitol)-induced [Ca^2+^]_i_ increase, and LaCl_3_ showed no effect on the [Ca^2+^]_i_ increase (Figure 7B). Similarly, only GdCl_3_ had a mild but significant effect on the NaCl (0.3 M)-induced [Ca^2+^]_i_ increase (Figure 7C). For the H_2_O_2_ (5 mM)-induced [Ca^2+^]_i_ increase, both 1 mM GdCl_3_ and 1 mM LaCl_3_ had a significant effect, and GdCl_3_ had a significant dose-dependent effect on the H_2_O_2_-induced [Ca^2+^]_i_ increase (Figure 7D).

These results indicated that different sources of Ca^2+^ are used in the calcium signaling responses induced by cold, drought, NaCl, and H_2_O_2_ stress.

### 2.8. Saline Environment Impairs the Calcium Signaling in Response to Various Stresses in P. patens

Long-term salinity acclimation in *P. patens* elicited a modulation of basal [Ca^2+^]_i_ levels. Overall, the basal [Ca^2+^]_i_ decreased in the saline environment. In the media containing 0–50 mM NaCl, the basal [Ca^2+^]_i_ decreased with increasing NaCl concentration, and when the concentration was increased to 60 and 70 mM NaCl, the basal [Ca^2+^]_i_ increased slightly, but was not significantly different from that at 50 mM. At 80 mM NaCl, the mean fluorescence value was 0.94 × 10^−2^, and it remained essentially unchanged as the concentration continued to increase (Appendix A). The concentration of NaCl corresponding to half of the basal [Ca^2+^]_i_ maximum was 20 mM, which we used as the optimum salt concentration for subsequent long-term salinity stress experiments.

The fluorescence (*L*/*L_max_*) measurements revealed that cold stress triggered [Ca^2+^]_i_ increases in 20 mM NaCl medium, showing a response pattern similar to NaCl-free conditions. However, *P. patens* grown under salt stress showed a significantly lower cold-induced [Ca^2+^]_i_ increase at 0–10 °C compared to NaCl-free controls (Figure 8A). The drought-induced [Ca^2+^]_i_ response in *P. patens* followed a concentration-dependent pattern under long-term saline conditions, similarly to the response observed in NaCl-free medium. While low mannitol concentrations (0.15 M and 0.4 M) elicited comparable [Ca^2+^]_i_ elevations in both saline and control environments, significant divergence emerged at higher osmotic stress levels (>0.5 M) (Figure 8B). *P. patens* exhibited attenuated [Ca^2+^]_i_ responses to H_2_O_2_ under saline conditions across all tested concentrations compared to NaCl-free controls. While the overall trend of oxidative stress-induced calcium signaling remained consistent between conditions, salt-adapted plants showed significantly reduced [Ca^2+^]_i_ elevation magnitudes (Figure 8C).

Finally, we treated *P. patens* grown in a saline environment with different concentrations of NaCl. The fluorescence values showed that high NaCl concentrations (≥0.4 M) induced comparable [Ca^2+^]_i_ responses in both salt-adapted and non-adapted plants, while significant differences emerged at lower concentrations (0.1–0.3 M), suggesting threshold-dependent signaling modulation by chronic salt stress (Figure 8D).

## 3. Discussion

Since the first transfer of apoaequorin cDNA into *Nicotiana plumbaginifolia* and its interaction with coelenterazine to reconstitute aequorin [25], aequorin has become an effective tool to study calcium signaling-mediated plant responses to various stresses [26]. In the present study, we improved the expression system and used the endogenous promoter *PpEF1-α* instead of the *CaMV35S* promoter to drive the stable expression of apoaequorin in *P. patens*. *PpEF1-α* promoter was considered to be more suitable for constructive expression in moss [27]. In addition to aequorin, other GECIs have also been used for Ca^2+^ detection, such as fluorescence resonance energy transfer (FRET) using Yellow Cameleon 2.1 (YC2.1) [28], YC2.12 [29], or the ultra-sensitive YC-Nano series [30]; however, these FRET-based Ca^2+^ indicators are complex to handle and have small measurement ranges. In addition, single-fluorescent calcium indicators, such as Camgaroos and GCaMPs, have also been used to detect changes in [Ca^2+^]_i_ [31,32,33]. Although these fluorescence-based GECIs have high spatial and temporal resolution, they require exogenous excitation to produce a signal, whereas bioluminescence-based GECIs do not require exogenous excitation and emit light as long as the substrate is present, avoiding problems such as toxicity due to light exposure. We used a highly sensitive CCD camera to directly detect the bioluminescence. Thus, the field of view is large enough to compare the different samples in one treatment, and it is suitable to carry out a large scale screening experiment in Ca^2+^ signaling. Also, there is sufficient space between the samples and the camera, and it is convenient for us to treat the samples during experiment. Due to these advantages, we chose aequorin-based Ca^2+^ imaging to reflect the Ca^2+^ signaling characteristics in *P. patens*. In fact, by using this system, several external stimulus receptors were previously discovered in *Arabidopsis* [34,35,36].

The cold, drought, salt, and oxidative stressors used in this study represent environmental factors that seriously affect plant growth and development. Plants increase their [Ca^2+^]_i_ upon sensing these abiotic stresses, as this process has been studied in depth in *Arabidopsis* and the related protein receptors have been screened [34,35,36]. In contrast, in bryophytes, a comprehensive investigation of the effect of these environmental stresses on calcium signaling using aequorin imaging is still lacking.

Whereas, in rice, the luminescence signal of aequorin can only be detected in the roots [9], *P. patens* has the advantage that the luminescence signal can be detected in the entire plant, which lacks specialized angiosperm structures such as roots, stems, or leaves. Although its chloroplasts produce chloroplast autofluorescence, our experimental preparation and manipulation phases were carried out in a dark environment, which greatly reduced the intensity of chloroplast autofluorescence so that it had little effect on the detection of changes in the calcium signal [9].

In *Arabidopsis*, the existing Ca^2+^ in the cells can be completely released using a discharging solution containing 10% ethanol [34]. In rice, a discharging solution of 25% ethanol was used [9]. In our study, the discharging solution was optimized for *P. patens*, and a discharging solution containing 15% ethanol was selected. This is because, from a histological point of view, the internal structure of the bryophytes is simple, and the epidermis, cortex, endodermis, and central cylinder of *Arabidopsis* are single-cell layers; in contrast, rice roots have an epidermis, a stromal tissue consisting of four tissues (exodermis, sclerenchyma cell layer, midcortex or mesodermis, and endodermis), and a central cylinder [37,38]. Rice roots, being more complex than *Arabidopsis* roots, require a greater ethanol concentration than *Arabidopsis* or *P. patens* to disrupt their cellular structure.

Low-temperature signaling in plants is dependent on Ca^2+^ signaling. Two Ca^2+^ channels, MID1-COMPLEMENTING ACTIVITY 1 (MAC1) and MAC2, were shown to interact and play a role in the low-temperature-induced transient Ca^2+^ increase in *Arabidopsis* [39]. CHILLING-TOLERANCE DIVERGENCE 1 (COLD1), a transmembrane protein located at the plasma membrane and ER, interacts with cold-responsive Ca^2+^ channels, thus playing a crucial role in the response to low temperature in rice [40]. We investigated the [Ca^2+^]_i_ response induced by different levels of cold stress and determined that 10–20 °C did not induce significant calcium signaling, whereas 0–5 °C induced a rapid and significant increase in calcium signals (Figure 3). Similarly, low-temperature treatment at 0–4 °C could significantly induce calcium signaling responses in both rice and *Arabidopsis* [26,40]. This indicated similar sensing mechanisms among these plant species. In addition, compared with previous research, our research provided more information about the kinetics of the [Ca^2+^]_i_ changes in *P. patens* [23].

Plants sense osmotic stress mainly through the hyperosmolality-gated Ca^2+^-permeable channel REDUCED HYPEROSMOLALITY-INDUCED Ca^2+^ INCREASE 1 (OSCA1), which in turn transmits signals downstream and plays a role in Ca^2+^-mediated osmotic signaling, prompting stomatal closure, plant wilting, and the inhibition of root growth [34]. In addition, CALCIUM PERMEABLE STRESS-GATED CATION CHANNEL 1 (CSC1) is also a Ca^2+^ channel regulated by osmotic stress [41]. For *P. patens*, drought stress had similar effects to cold stress in that low concentrations of mannitol had less of an effect on the increase in [Ca^2+^]_i_, whereas higher concentrations of mannitol induced a rapid increase in [Ca^2+^]_i_ (Figure 4). These findings are similar to those obtained using YC3.60 to detect mannitol-induced calcium signals in *P. patens* [42]. However, when comparing the calcium signaling responses to osmotic stress between *P. patens* and *Arabidopsis*, we found that *P. patens* exhibited a less sensitive calcium signaling response under drought conditions. According to previous studies, 200 mM mannitol could significantly induce calcium signaling in *Arabidopsis* [26]. However, we found that even 400 mM mannitol failed to trigger calcium signaling in *P. patens*. As the pioneer plant species that first invaded land ecosystems, the insensitivity of *P. patens* in response to osmotic stress may indicate their distinct evolutionary traits, stress response pathways, cellular architecture, and ecological adaptation strategies.

In contrast to its insensitivity to drought, *P. patens* exhibited high sensitivity to salt stress, with 0.1 M NaCl treatment sufficient to trigger significant changes in calcium signaling. Similar results were observed in *Arabidopsis* [26]. However, in rice, 0.1 M NaCl treatment failed to induce noticeable calcium signaling changes [9]. Investigating the molecular mechanisms behind these varying salt stress sensitivities across different plant species will provide deeper insights into plant salt stress adaptation mechanisms. The 2 M NaCl treatment induced a very large fluorescence value, indicating that high-concentration NaCl may have had adverse effects on the cell structure of *P. patens*, preventing the moss from recovering to its basal state. Thus, before the discharging solution treatment, most of the Ca^2+^ had already been lost, meaning that *L_max_* did not reflect all the intracellular Ca^2+^. These results are very similar to those for the response of rice to salt stress [9].

ROS are associated with plant responses to stimuli, such as drought, salt, and low temperatures, and integrate stress signals [18]. Extracellular H_2_O_2_ (eH_2_O_2_) was shown to increase [Ca^2+^]_i_ in *N. plumbaginifolia* [43]. The eH_2_O_2_-activated Ca^2+^ channels mediate the Ca^2+^ influx into protoplasts and cause an increase in the intact guard cell [Ca^2+^]_i_ [44]. In *Arabidopsis*, the eH_2_O_2_-induced increase in [Ca^2+^]_i_ is mediated by the cell surface receptor HYDROGEN-PEROXIDE-INDUCED Ca^2+^ INCREASES 1 (HPCA1) [36]. Here, we showed that oxidative treatments induced a sensitive reaction similar to salt stress, with significant calcium signaling induced by low concentrations of H_2_O_2_, suggesting that *P. patens* is sensitive to ROS. This highly sensitive calcium signaling response to ROS was also observed in *Arabidopsis* and rice [9,26]. Given the role of ROS as crucial signaling molecules in plants [18], we propose that ROS plays an important role in stress resistance mechanisms across various plant species. Despite these commonalities, there are subtle differences in calcium signaling responses to ROS stress across different plant species. In *Arabidopsis*, a second peak of the luminescence signal occurred 5–20 min after H_2_O_2_ treatment [26]. However, this phenomenon was not observed in rice or *P. patens*, and the underlying reasons remain to be further investigated [9].

Various environmental factors affect the Ca^2+^ channels in plants, thereby inducing a transient increase in [Ca^2+^]_i_ [18]. The results of the present study showed that sufficient concentrations of GdCl_3_, LaCl_3_, neomycin, and thapsigargin almost completely inhibited the increase in [Ca^2+^]_i_ in response to low temperatures. It is worth noting that the inhibitory effect of GdCl_3_ and LaCl_3_ was different for the mannitol-, NaCl-, and H_2_O_2_-induced [Ca^2+^]_i_ responses, which suggest different properties for these two similar chemical inhibitors. All these results suggest that *P. patens* induces an increase in [Ca^2+^]_i_ through different channels under different environmental stimuli.

In addition, consistent with the different sensitivities of *P. patens* and rice to NaCl treatment, the effects of inhibitors on were also different between these two species. In rice, all four inhibitors were able to suppress the NaCl-induced [Ca^2+^]_i_ response to varying degrees, whereas in *P. patens*, only GdCl_3_ showed an inhibitory effect [9]. This indicated that the Ca^2+^ channels involved in NaCl-induced [Ca^2+^]_i_ response are different between rice and *P. patens*. In contrast to the varying effects of different inhibitors on the NaCl-induced [Ca^2+^]_i_ response in rice and *P. patens*, their effects on H_2_O_2_-induced [Ca^2+^]_i_ response are quite similar. Both GdCl_3_ and LaCl_3_ showed significant inhibitory effects in rice and *P. patens*, while neomycin and thapsigargin had minimal effects. Considering the similar sensitivity of calcium signaling response to H_2_O_2_, we suggest that the Ca^2+^ channels involved in the H_2_O_2_-induced [Ca^2+^]_i_ response may be the same in rice and moss.

Plants in the wild environment are often subjected to various combined stresses. Thus, we examined whether the response of plants to other stresses is altered under long-term salt stress environments, and observed that higher-magnitude changes in [Ca^2+^]_i_ occur with increasing stress intensity, as was previously observed in *Arabidopsis* [45]. Although cold, drought, oxidative stress, and additional NaCl treatments still affected the [Ca^2+^]_i_ of *P. patens* grown under prolonged salt stress, we observed that the moss was less sensitive to other stimuli than when grown under normal conditions. This phenomenon has been demonstrated in physiological experiments in other species [46,47,48]. Alternatively, because prolonged salt stress decreases the basal [Ca^2+^]_i_ of *P. patens*, the [Ca^2+^]_i_ increases under additional stimuli may have been of similar magnitude, although lower in their absolute values. This phenomenon therefore requires further elucidation in more detailed experiments.

## 4. Materials and Methods

### 4.1. Plant Materials and Growth Conditions

*P. patens* ‘Gransen’ was the wild-type (WT) strain. The moss was cultured on BCDATG medium with 0.5% (*w*/*v*) sucrose and 0.8% (*w*/*v*) agar at 25 °C under a 16 h light–8 h dark photoperiod and a light intensity of 50–80 µmol photons m^−2^ s^−1^. Protonemata of *P. patens* were grown on BCDATG medium under continuous white light [49].

### 4.2. Vector Construction and Transformation of P. patens

In order to generate *P. patens* efficiently expressing the intracellular Ca^2+^ indicator aequorin, the pTN182 vector plasmid was the basic skeleton. In order to search for genomic regions for exogenous DNA fragment insertion in *P. patens*, the PIG1 site was selected as previously described [50]. The upstream homologous arm, PIG1bL (949 bp), was amplified via PCR from the genomic DNA of *P. patens* using primers Left-F and Left-R, and cloned into the *Bam*HI–*Sac*I sites of the pTN182 plasmid; the resulting plasmid was named pTN182-LEFT. The apoaequorin cDNA along with the pea *rbcS* terminator were amplified via PCR from pCAMBIA1300-AQ [9] using the primers AQ-F and AQ-R, and cloned into the *Sal*I–*Eco*RI sites of the pTN182-LEFT plasmid; the resulting plasmid was named pTN182-AQ-LEFT. The downstream homologous arm PIG1bR (1029 bp) and the endogenous *PpEF1-α* promoter were amplified via PCR from pPOG1 using the primers Right-P-F and Right-P-R, and cloned into the *Kpn*I–*Xho*I sites of pTN182-AQ-LEFT; the resulting plasmid was named pTN182-RIGHT-AQ-LEFT. The DNA fragments for transformation via homologous recombination were amplified via PCR from the aequorin expression vector pTN182-RIGHT-AQ-LEFT using the primers Right-F and Left-R. The primers used to construct the plasmids and prepare the transformation fragment are listed in Appendix A.

The transformation was performed using the polyethylene glycol (PEG)-mediated method [51]. Briefly, 30 µg (20 µL) DNA fragments was added to the resuspension containing 5 × 10^5^ protoplasts and treated with PEG. Stably transformed strains were obtained by two successive cycles of incubation on a selective medium (the concentrations of G418 were 25 µg mL^−1^ and 50 µg mL^−1^, respectively) and nonselective medium. The surviving plants were verified using genomic PCR and RT-PCR. The primers used for the transformation validation are listed in Appendix A.

### 4.3. Aequorin Bioluminescence–Based Ca^2+^ Imaging

The [Ca^2+^]_i_ was measured in *P. patens* expressing aequorin. The cormus of *P. patens* was grown on BCDATG medium in a 90 mm diameter round Petri dish for seven to nine days. A treatment of 250 µL of 10 µM coelenterazine (Prolume) was evenly applied to the plant materials 6–12 h before imaging and placed in the dark to reconstitute aequorin in vivo. The aequorin bioluminescence imaging was performed using a ChemiPro HT system (Roper Scientific, Trenton, NJ, USA), as described previously [34]. The luminescence images were analyzed using WinView/32 (v2.5.16, Roper Scientific) and ImageJ (v1.53k, https://imagej.net/ij/, accessed on 15 March 2024). The luminescence counts (*L*) [34,35] were recorded continuously for 3–5 min after treatment with the different abiotic stresses. For the optimization of the discharging solution in *P. patens*, we modified the discharging solution previously used for *Arabidopsis* (0.9 M CaCl_2_ and 10% anhydrous ethanol (*v*/*v*)) and rice (0.9 M CaCl_2_ and 25% anhydrous ethanol (*v*/*v*)) [9], and established a series of ethanol concentration gradients (0%, 10%, 15%, 20%, 25%, 30%, 40%, and 50%). *P. patens* was treated with discharging solutions containing different ethanol concentration gradients, and the luminescence signals were measured. The ethanol concentration that produced the highest luminescence intensity was determined as the optimal concentration for the discharging solution in this study.

To investigate the [Ca^2+^]_i_ changes in *P. patens* in response to cold stress, pure water with different temperatures was used in the application of low temperature, and the luminescence of aequorin-based Ca^2+^ imaging under different temperature treatments was detected and compared. To study the changes of [Ca^2+^]_i_ in response to drought, salt and ROS stresses, the solutions of varying concentrations of mannitol, NaCl and H_2_O_2_ were used to subject the *P. patens* to different stress treatments. The luminescence of aequorin-based Ca^2+^ imaging under different stresses was detected and compared. To investigate the dynamic process of [Ca^2+^]_i_ induction by different stresses in *P. patens* over time, the plants were subjected to stress treatment at 0 s, the luminescence images were taken every 15 s after the treatment, and the images were taken continuously for 3–5 min. The chloroplast autofluorescence images were recorded continuously for 2 min after treatment with strong light for 30–60 s to determine the growth status and positioning of the plants. The total remaining aequorin luminescence counts (*L_max_*) [34,35] were recorded continuously for 3 min after treatment with the Ca^2+^ discharging solution to release all intracellular Ca^2+^. All experiments were conducted in the dark. In this experiment, the pure water of room temperature was the control to exclude the interference of the mechanical stimulation of *P. patens*.

### 4.4. Ca^2+^ Source Analysis

To investigate the sources of Ca^2+^ in the cold-, drought-, NaCl-, and H_2_O_2_- induced [Ca^2+^]_i_ increases, we examined the [Ca^2+^]_i_ increase in response to these abiotic stresses in the plants treated with different Ca^2+^ channel inhibitors. Plant materials were treated with different concentrations of the Ca^2+^ inhibitors GdCl_3_, LaCl_3_, neomycin, or thapsigargin for 1 h in the dark, after which they were subjected to a treatment consisting of 10 °C, 0.5 M mannitol, 0.3 M NaCl, and 5 mM H_2_O_2_. The *L*, *L_max_*, and chloroplast autofluorescence images were subsequently recorded.

### 4.5. Various Stress Treatments Under Long-Term Salt Stress

To analyze the effect of long-term salinity on the [Ca^2+^]_i_ response of *P. patens* in response to various stresses, we first determined an appropriate salt environment for the culture of this species. *P. patens* was cultured on BCDATG medium supplemented with NaCl at 10 mM intervals, ranging from 0 to 100 mM, to simulate various salinities and incubated for 10 days, after which *L* was recorded for 30 min. The plants were then treated with discharging solution, and the luminescence signal was collected for 3 min. The fluorescence value (*L*/*L_max_*) was calculated to represent the basal [Ca^2+^]_i_. The NaCl concentration in the medium corresponding to half of the normal basal [Ca^2+^]_i_, i.e., the *K*_d_ value, was chosen as the optimum salt concentration. For the various stress treatments under long-term salt stress, solutions of varying concentrations were used to treat the *P. patens* grown on BCDATG medium with specific concentration of NaCl (based on the calculations, the concentration was ultimately determined to be 20 mM).

### 4.6. Statistical Analyses

For Figure 2B, Figure 3B, Figure 4B, Figure 5B and Figure 6B, one-way ANOVA was performed, followed by Tukey’s honestly significant difference (HSD) post hoc test (α = 0.05). To examine the effects of inhibitor type and concentration on the treatment outcomes, as well as their potential interaction (Figure 7), a two-way ANOVA was performed, followed by Tukey’s HSD post hoc test to identify significant differences among groups. The results of the multiple comparisons were represented using Tukey’s letters, where groups sharing the same letter indicate no significant difference, while different letters denote significant differences. To evaluate the significant differences between the long term NaCl-treated and control groups (Figure 8), the multiple regression analysis with categorical analyses were performed. Significant effects were further analyzed using Tukey’s HSD post hoc test, and results were represented using asterisks to denote significant differences. All statistical analyses were performed using Origin, and a significance level of *p* < 0.05 was applied.

## 5. Conclusions

In this research, we transferred the apoaequorin cDNA into *P. patens* and reconstituted aequorin in vitro. The ethanol concentration in the discharging solution was optimized to 15%, which was different from the concentrations used for rice and *Arabidopsis*. Next, the [Ca^2+^]_i_ changes induced by cold, drought, salt, and oxidative stresses were measured and compared. In addition, the dynamic process of [Ca^2+^]_i_ induction by different abiotic stresses over time was also investigated. The results showed that the characteristics of cold- and H_2_O_2_-induced [Ca^2+^]_i_ changes in *P. patens* were similar to those in rice and *Arabidopsis*, while drought- and NaCl-induced [Ca^2+^]_i_ changes in *P. patens* exhibited some differences compared to rice or *Arabidopsis*. This indicated the different Ca^2+^ channels involved in these processes in *P. patens*, which was partially proved by the different effects of Ca^2+^ channel inhibitors. Finally, the effect of long-term salinity on the [Ca^2+^]_i_ response of *P. patens* to various stresses was evaluated, and we observed that long-term salinity suppressed the basal [Ca^2+^]_i_ of *P. patens*, and the peak value of [Ca^2+^]_i_ was lower than that of plants growing in non-stressed environments. This is the first systematic study of calcium signaling in bryophytes, and our research provided an efficient and convenient tool to study calcium signaling in bryophytes.

While this aequorin bioluminescence-based system offers a powerful approach for monitoring whole-plant calcium signaling in *P. patens*, its camera-based detection limits resolution to the organismal level, precluding cellular-scale analysis. Nevertheless, this tool opens new possibilities for investigating stress signaling mechanisms in bryophytes. Given that many environmental sensors have been identified through similar approaches in *Arabidopsis* [34,35,36], future studies could explore functional homologs in *P. patens*, particularly focusing on the unique NaCl response pathways suggested by our inhibitor experiments. The distinct pharmacological profiles observed between *P. patens* and angiosperms imply that bryophytes may possess novel calcium signaling components, making this far from a repetitive endeavor. Such comparative studies will not only deepen our understanding of stress adaptation in early land plants but may also reveal the evolutionarily conserved and divergent elements of calcium signaling networks.

## Figures and Tables

**Figure 1 plants-14-01178-f001:**
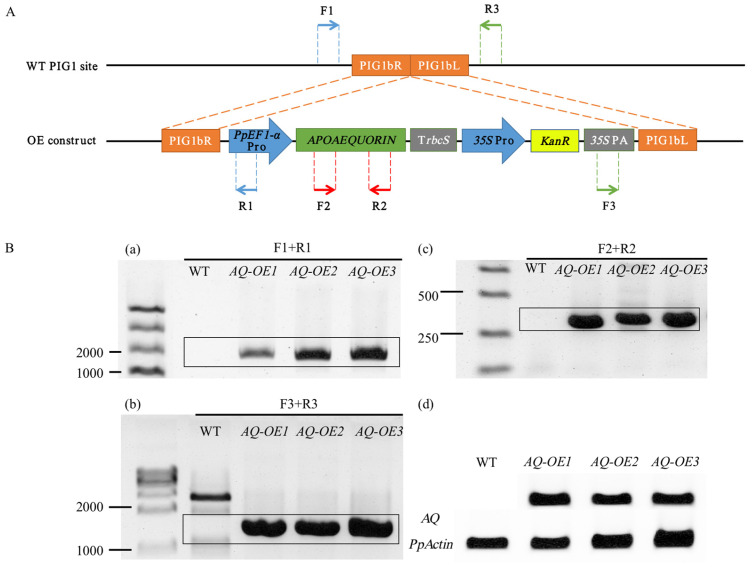
Recombinant expression of *APOAEQUORIN* in stable transgenic *P. patens* lines. (**A**) Schematic representation of the constructs for the expression of *APOAEQUORIN*. Arrows represent primers used in the PCR identification of transgenic lines. PIG1bR and PIG1bL are two adjacent DNA fragments of insertion site in *P. patens*. (**B**) Identification of transformants using a PCR analysis of genomic DNA isolated from the wild-type (WT) and transgenic lines and an RT-PCR detection of *APOAEQUORIN* expression in these plants. The PCR identification of the upstream homologous arm (a), downstream homologous arm (b), and *APOAEQUORIN* gene (c). RT-PCR detection of *APOAEQUORIN* expression (d).

**Figure 2 plants-14-01178-f002:**
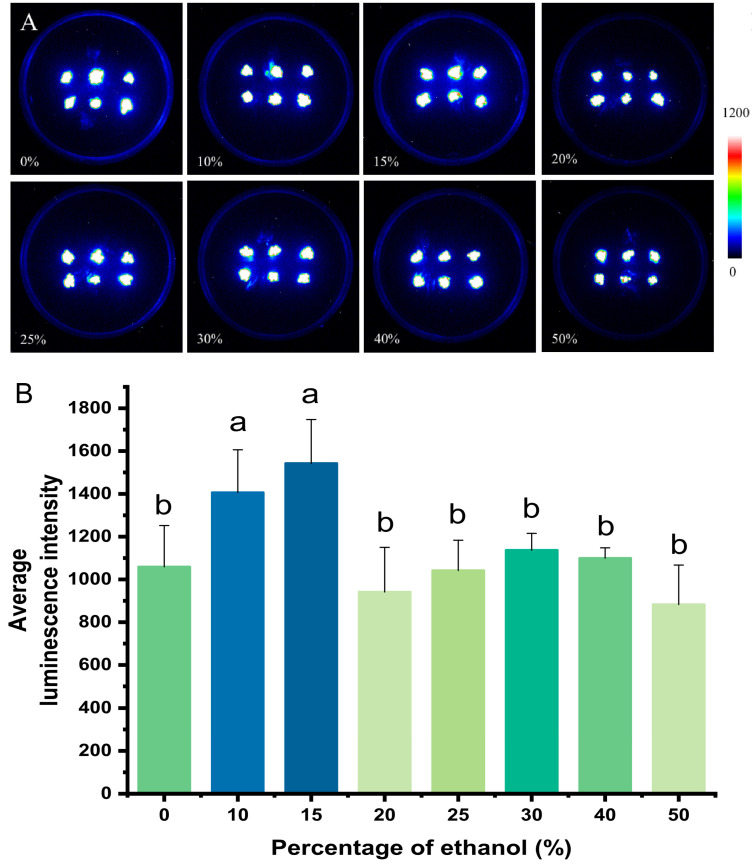
Optimization of the discharging solution for *P. patens*. (**A**) Pseudo-color images of aequorin luminescence in *P. patens* treated with discharging solutions containing different percentages of ethanol. The relationships between luminescence intensity and the pseudo-color images are scaled by a pseudo-color bar. (**B**) Average luminescence intensity of every treatment. Data for independent experiments are shown (mean ± sd, *n* = 12, a one-way analysis of variance (ANOVA) with post hoc analysis was conducted. The number “*n*” indicates the number of tested plates with six clusters of the mosses. Different letters indicate significant differences).

**Figure 3 plants-14-01178-f003:**
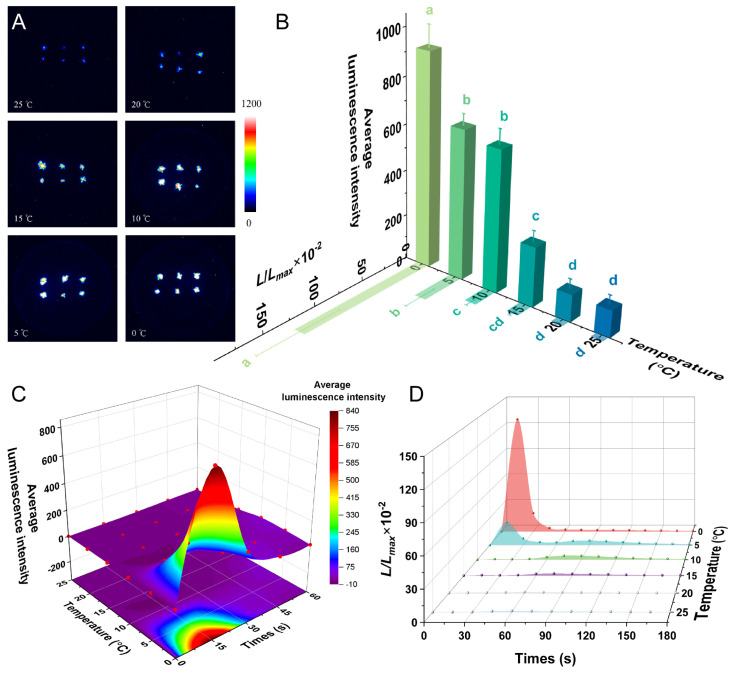
Calcium signaling induced by cold stress. (**A**) Pseudo-color images of aequorin luminescence in *P. patens* treated with different temperatures. The relationships between luminescence intensity and the pseudo-color images are scaled by a pseudo-color bar. (**B**) Average luminescence intensity and *L/L_max_* of every treatment. Data for independent experiments are shown (mean ± sd, *n* = 12, one-way ANOVA. Different letters indicate significant differences). (**C**) Time course of the average luminescence intensity following treatments with different temperatures (0–60 s). (**D**) Time course of the *L/L_max_* following treatments with different temperatures (0–180 s). Six replicates were performed for each treatment.

**Figure 4 plants-14-01178-f004:**
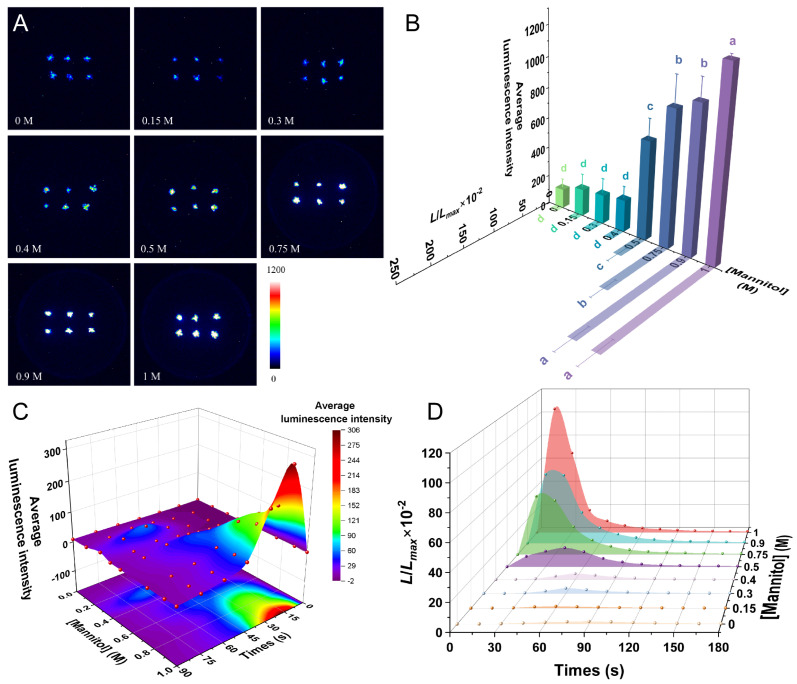
Calcium signaling induced by drought. (**A**) Pseudo-color images of aequorin luminescence in *P. patens* treated with different concentrations of mannitol. The relationships between luminescence intensity and the pseudo-color images are scaled by a pseudo-color bar. (**B**) Average luminescence intensity and *L/L_max_* of every treatment. Data for independent experiments are shown (mean ± sd, *n* = 12, one-way ANOVA. Different letters indicate significant differences). (**C**) Time course of the average luminescence intensity following treatments with different concentrations of mannitol (0–90 s). (**D**) Time course of the *L/L_max_* following treatments with different concentrations of mannitol (0–180 s). Six replicates were performed for each treatment.

**Figure 5 plants-14-01178-f005:**
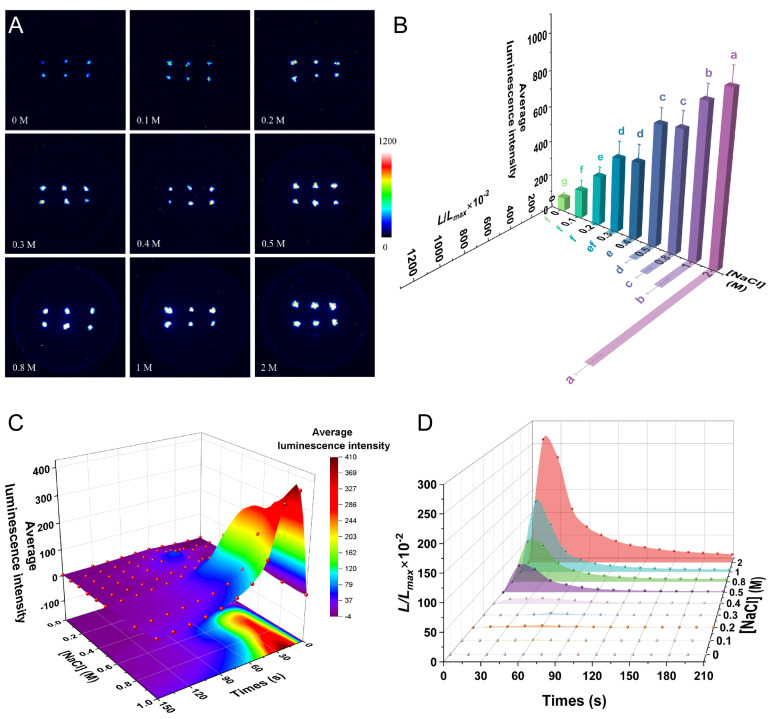
Calcium signaling induced by NaCl. (**A**) Pseudo-color images of aequorin luminescence in *P. patens* treated with different concentrations of NaCl. The relationships between luminescence intensity and the pseudo-color images are scaled by a pseudo-color bar. (**B**) Average luminescence intensity and *L/L_max_* of every treatment. Data for independent experiments are shown (mean ± sd, *n* = 12, one-way ANOVA. Different letters indicate significant differences). (**C**) Time course of the average luminescence intensity following treatments with different concentrations of NaCl (0–150 s). (**D**) Time course of the *L/L_max_* following treatments with different concentrations of NaCl (0–210 s). Six replicates were performed for each treatment.

**Figure 6 plants-14-01178-f006:**
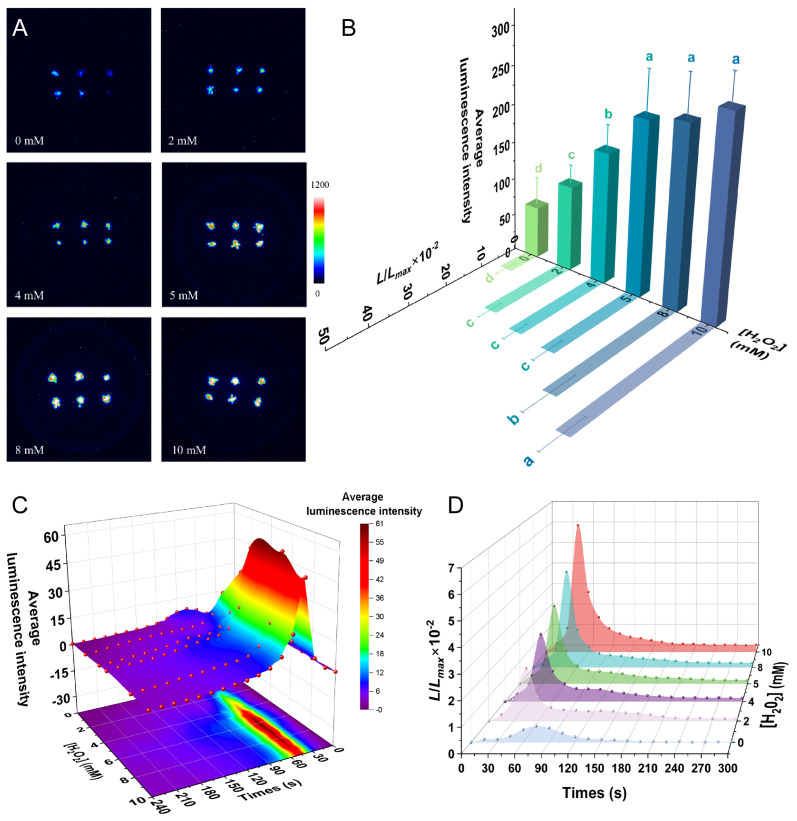
Calcium signaling induced by oxidative stress. (**A**) Pseudo-color images of aequorin luminescence in *P. patens* treated with different concentrations of H_2_O_2_. The relationships between luminescence intensity and the pseudo-color images are scaled by a pseudo-color bar. (**B**) Average luminescence intensity and *L/L_max_* of every treatment. Data for independent experiments are shown (mean ± sd, *n* = 12, one-way ANOVA. Different letters indicate significant differences). (**C**) Time course of the average luminescence intensity following treatments with different concentrations of H_2_O_2_ (0–240 s). (**D**) Time course of the *L/L_max_* following treatments with different concentrations of H_2_O_2_ (0–300 s). Six replicates were performed for each treatment.

**Figure 7 plants-14-01178-f007:**
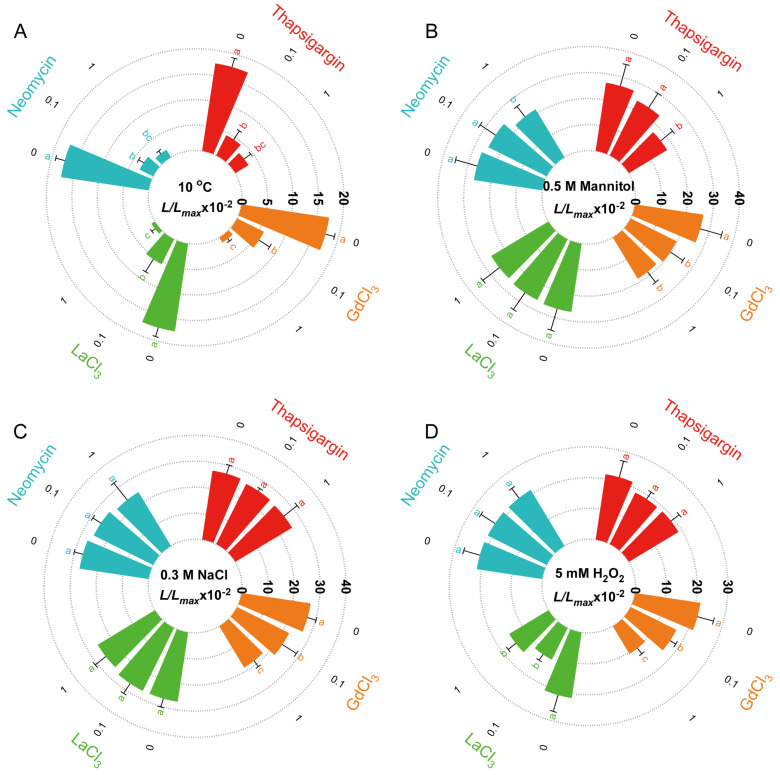
Effects of the different concentrations of Ca^2+^ inhibitors on the increase in [Ca^2+^]_i_. The effect of different concentrations of Ca^2+^ inhibitors on plants grown in abiotic stress conditions: 10 °C (**A**), 0.5 M mannitol solution (**B**), 0.3 M NaCl solution (**C**), and 5 mM H_2_O_2_ solution (**D**). The concentration gradients for inhibitors 0.1 mM, and 1 mM. Data for independent experiments are shown (mean ± sd, *n* = 12, one-way ANOVA. Different letters indicate significant differences).

**Figure 8 plants-14-01178-f008:**
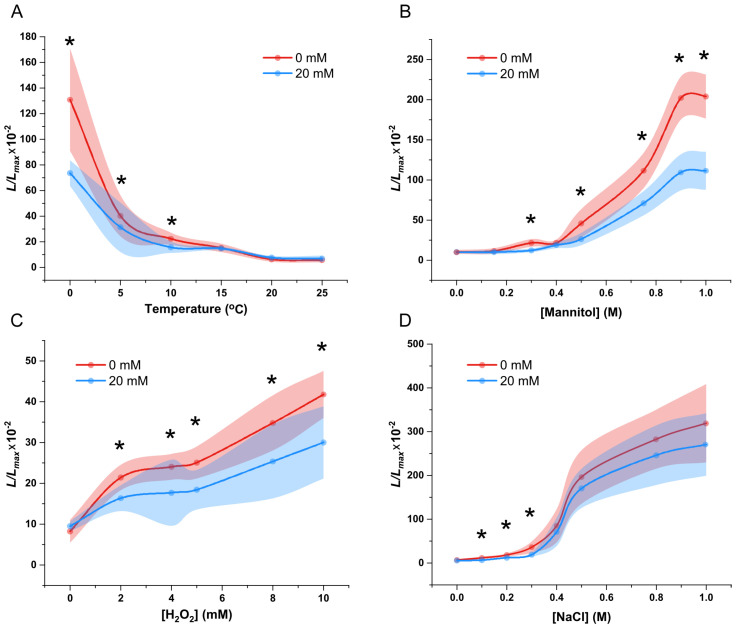
Calcium signaling induced by cold, drought, oxidative, and salt stresses in plants grown under prolonged salt stress. The *L/L_max_* of *P. patens* in response to cold (**A**), drought (**B**), oxidative (**C**), and additional NaCl (**D**) stresses under salt-stressed and normal environments (mean ± sd, *n* = 12. Asterisks indicate significant differences).

## Data Availability

The original contributions presented in this study are included in the article/Appendix A. Further inquiries can be directed to the corresponding authors.

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
