# Peer review of "Aquorin Bioluminescence-Based Ca2+ Imaging Reveals Differential Calcium Signaling Responses to Abiotic Stresses in Physcomitrella patens"

_plants, 2025, doi:10.3390/plants14081178_

Round 1

Reviewer 1 Report

Comments and Suggestions for Authors

In this paper, the authors studied aequorin bioluminescence-based Ca2+ imaging to reveal the differential calcium signaling responses to abiotic stresses in Physcomitrella patens. They proposed a novel technique to detect the stress reaction under different conditions. A lot of data was collected. However, the data analysis was poor and lacked a theoretical background. In this stage, the paper could not be recognized as a research paper. Please ask for help from a statistical expert to analyze these precious data. The statistical technique and data analysis method should be supplied in section 4. Materials and Methods. After finishing the data analysis, the Results and Discussion part should be written.

  1. In Figure 2, the authors mentioned using the one-way analysis of variance. Was the post hoc method applied to produce the (b) part?
  2. In Figure 2, why is the average luminescence intensity of anhydrous ethanol percentages lower than that of the 15% and 25%? Please explain the irregular reaction.
  3. In Figure 3, “Calcium signaling induced by cold stress,” There are two factors: time and temperature. Why not use two-way ANOVA?
  4. The same question in Figures 4, 5, and 6: Why not use two-way ANOVA?
  5. In Figure 7. “Effect of different concentrations of Ca2+ inhibitors on the increase of [Ca2+]i. “There are two influencing factors, so the two-way ANOVA should be used.
  6. In Figure 8, please use the multiple regression and categorical analyses to evaluate the significant difference between factors. The results presented in this figure were inadequate.

Author Response

In this paper, the authors studied aequorin bioluminescence-based Ca2+ imaging to reveal the differential calcium signaling responses to abiotic stresses in Physcomitrella patens. They proposed a novel technique to detect the stress reaction under different conditions. A lot of data was collected. However, the data analysis was poor and lacked a theoretical background. In this stage, the paper could not be recognized as a research paper. Please ask for help from a statistical expert to analyze these precious data. The statistical technique and data analysis method should be supplied in section 4. Materials and Methods. After finishing the data analysis, the Results and Discussion part should be written.

Response: We have re-analyzed the data in accordance with the reviewers' comments and documented the methodology in the Materials and Methods section. In addition, some parts of Results and Discussion were re-written accordingly.

1. In Figure 2, the authors mentioned using the one-way analysis of variance. Was the post hoc method applied to produce the (b) part?

Response: Yes, we were not aware in advance which concentration of ethanol would be the most suitable, so we employed a post hoc analysis method. We have made the corresponding revisions in the figure legend of Figure 2B.

2. In Figure 2, why is the average luminescence intensity of anhydrous ethanol percentages lower than that of the 15% and 25%? Please explain the irregular reaction.

Response: The role of the discharging solution is to release all intracellular Ca2+ and allow all aquorin to exhibit luminescence. Thus, it is important to optimize the ethanol concentration in the discharging solution. As the ethanol concentration in the solution increases, cell damage intensifies, enabling more aquorin to emit luminescence. However, with further increases in ethanol concentration, ethanol itself may begin to inhibit the luminescence response of aquorin. As a result, we observe that the luminescence intensity initially increases with rising ethanol concentration but decreases after the ethanol concentration exceeds 15%.

 3. In Figure 3, “Calcium signaling induced by cold stress,” There are two factors: time and temperature. Why not use two-way ANOVA?

Response: The statistical analysis and comparisons were limited to Figure 3B, where only one factor—temperature—was involved. Therefore, we used one-way ANOVA.

4. The same question in Figures 4, 5, and 6: Why not use two-way ANOVA?

Response: The same with Figures 4, 5, and 6. The statistical analysis and comparisons were limited to Figure 4B, 5B and 6B, where only one factor was involved. Therefore, we used one-way ANOVA.

5. In Figure 7. “Effect of different concentrations of Ca2+ inhibitors on the increase of [Ca2+]i. “There are two influencing factors, so the two-way ANOVA should be used.

Response: We have revised the Figure 7 using two-way ANOVA as suggested.

6. In Figure 8, please use the multiple regression and categorical analyses to evaluate the significant difference between factors. The results presented in this figure were inadequate.

Response: We have revised the Figure 7 using the multiple regression and categorical analyses as suggested.

Reviewer 2 Report

Comments and Suggestions for Authors

 Dear Authors,

I have carefully reviewed your manuscript. While the study presents interesting findings, several improvements are necessary to enhance its accuracy, clarity, and overall quality. Please find my detailed comments below:

1. Lines 59–64: This paragraph is more suitable for the conclusion section. I recommend removing it from here and integrating it into the discussion and/or conclusion. Instead, please provide a clear hypothesis for your study in this section.

2. Figure 2B: For better clarity in comparisons, I recommend using asterisks or Tukey’s letters instead of p-values, which may be confusing to readers.

3. Line 113: If a treatment does not show a statistically significant effect on a particular parameter, there is no need to report the p-value. Please check the manuscript thoroughly for similar instances and revise accordingly.

4. Lines 122–133: Avoid using uncommon abbreviations. I suggest either writing out the full terms or including the full form in parentheses to ensure clarity and facilitate understanding of the procedures and results.

5. Section 2.5 (NaCl Induces Calcium Ion Spikes in P. patens): How were the salt stress treatments selected? Were they based on previous studies? If so, please provide appropriate citations. Additionally, I am curious whether the plants survived under 1–5 M NaCl. Please clarify.

6. Lines 246–248: The information regarding rice appears irrelevant to this study, as it pertains to a completely different crop. However, if this information is essential, it would be more appropriate to include it in the introduction as part of the literature review.

7. Figure 8: Please indicate statistical differences using either asterisks or Tukey’s letters for clarity.

8. Discussion Section: The discussion should be rewritten to provide a stronger analytical perspective. I recommend a structured comparison of your findings with previous studies, highlighting both similarities and differences. The similarities will reinforce your results, while the differences will emphasize the novelty and significance of your research.

9. Materials and Methods: This section requires significant improvement. It is unclear how different stress conditions were applied. For example, in line 521, it is mentioned that the medium contained 0–100 mM NaCl, yet later, concentrations of 1, 3, and 5 M NaCl are referenced. Please clarify how these concentrations were achieved. Additionally, include a subsection titled “Statistical Analyses” to explain all the statistical methods used in the study.

10. Additional Sections: I suggest adding two new sections to the manuscript:

• Future Perspectives: Discuss the potential implications of your findings and possible directions for future research.

• Limitations and Challenges: Address any constraints or challenges encountered in your study.

11. Visual Abstract: Please create a visual abstract illustrating the calcium signaling network and the possible effects of abiotic stressors. This will improve the manuscript’s readability and impact.

12. References: Carefully review and revise all references to ensure compliance with MDPI’s reference formatting guidelines.

13. I could not find a clear conclusion section in the manuscript, which is essential for summarizing the key findings and their implications. 

Author Response

I have carefully reviewed your manuscript. While the study presents interesting findings, several improvements are necessary to enhance its accuracy, clarity, and overall quality. Please find my detailed comments below:

  1. Lines 59–64: This paragraph is more suitable for the conclusion section. I recommend removing it from here and integrating it into the discussion and/or conclusion. Instead, please provide a clear hypothesis for your study in this section.

Response: We have removed this paragraph and integrated it into the conclusion. In addition, we have provided a  hypothesis (comparison of Calcium signaling between P. patens and the angiosperm models Arabidopsis and rice) in this section as suggested.

  1. Figure 2B: For better clarity in comparisons, I recommend using asterisks or Tukey’s letters instead of p-values, which may be confusing to readers.

Response: We have used asterisks instead of p-values as suggested.

  1. Line 113: If a treatment does not show a statistically significant effect on a particular parameter, there is no need to report the p-value. Please check the manuscript thoroughly for similar instances and revise accordingly.

Response: We have check the manuscript thoroughly for similar instances and revise accordingly as suggested.

  1. Lines 122–133: Avoid using uncommon abbreviations. I suggest either writing out the full terms or including the full form in parentheses to ensure clarity and facilitate understanding of the procedures and results.

Response: We included the full form in parentheses as suggested.

  1. Section 2.5 (NaCl Induces Calcium Ion Spikes in P. patens): How were the salt stress treatments selected? Were they based on previous studies? If so, please provide appropriate citations. Additionally, I am curious whether the plants survived under 1–5 M NaCl. Please clarify.

Response: The salt stress treatments were based on previous studies, and we have provided citations as suggested. Almost no plants are able to survive at such high levels of salinity (1–5 M). The reason we subjected the plants to such high salt concentrations was to identify the saturation point of calcium ion signaling, which has been observed in other plant species—specifically, the point beyond which calcium ion signals no longer increase with higher salt concentrations. However, in P. patens, we did not observe such a saturation point. The calcium ion signal continued to amplify even at a salt concentration of 1 M. When the salt concentration reached 2 M, we observed damage to the P. patens tissue, and therefore, we did not proceed with treatments at higher salt concentrations.

  1. Lines 246–248: The information regarding rice appears irrelevant to this study, as it pertains to a completely different crop. However, if this information is essential, it would be more appropriate to include it in the introduction as part of the literature review.

Response: We have removed this sentence as suggested.

  1. Figure 8: Please indicate statistical differences using either asterisks or Tukey’s letters for clarity.

Response: We have used asterisks to indicate statistical differences as suggested.

  1. Discussion Section: The discussion should be rewritten to provide a stronger analytical perspective. I recommend a structured comparison of your findings with previous studies, highlighting both similarities and differences. The similarities will reinforce your results, while the differences will emphasize the novelty and significance of your research.

Response: We have revised the discussion section to emphasize both the similarities and differences in calcium signaling responses to cold, drought, salt, ROS, and inhibitors between our findings and previously reported results as suggested.

  1. Materials and Methods: This section requires significant improvement. It is unclear how different stress conditions were applied. For example, in line 521, it is mentioned that the medium contained 0–100 mM NaCl, yet later, concentrations of 1, 3, and 5 M NaCl are referenced. Please clarify how these concentrations were achieved. Additionally, include a subsection titled “Statistical Analyses” to explain all the statistical methods used in the study.

Response: We have revised the Materials and Methods section to ensure that readers can clearly understand the procedures for various stress treatments. In addition, the subsection of “Statistical Analyses” has been included as suggested.

  1. Additional Sections: I suggest adding two new sections to the manuscript:
  • Future Perspectives: Discuss the potential implications of your findings and possible directions for future research.
  • Limitations and Challenges: Address any constraints or challenges encountered in your study.

 Response: We have added these two new sections to the manuscript as suggested.

  1. Visual Abstract: Please create a visual abstract illustrating the calcium signaling network and the possible effects of abiotic stressors. This will improve the manuscript’s readability and impact.

Response: We have created a visual abstract (Graphical Abstract) as suggested.

  1. References: Carefully review and revise all references to ensure compliance with MDPI’s reference formatting guidelines.

Response: We have reviewed and revised all references as suggested.

  1. I could not find a clear conclusion section in the manuscript, which is essential for summarizing the key findings and their implications. 

Response: We have created the Conclusion section as suggested.

Reviewer 3 Report

Comments and Suggestions for Authors

The work has scientific relevance, but some points will need to be reviewed. The following are the observations:

Keywords should not repeat words in the title

Specify the abiotic stresses studied in the introduction

The introduction should present the objective and hypothesis of the work and not the conclusion.

Item 2.2, we suggest inserting the result first and then the corresponding figure

Do not include in results what is part of the methodology _lines 93-99; 108-109; 121-123; 143-144; 163; 181-183; 204-205; 238-239, 262-265; 288-290, 298-300; 307-310; 319-321.

Lines 246-248: Wouldn't this paragraph be more appropriate in discussion?

Line 402: More important than comparing is explaining why this happened.

Check some words are in upper case in the discussion.

At the end of the discussion, the future perspective could simply be included after the results obtained.

Line 465:The description of the experiments with stresses was missing

Include the conclusion item

Author Response

The work has scientific relevance, but some points will need to be reviewed. The following are the observations:

Keywords should not repeat words in the title

Response: We have changed the keywords and ensured that they do not appear in the title.

Specify the abiotic stresses studied in the introduction

Response: We have specified the abiotic stresses in the introduction as suggested.

The introduction should present the objective and hypothesis of the work and not the conclusion.

Response: We have presented the objective and hypothesis of the work in the end of introduction as suggested.

Item 2.2, we suggest inserting the result first and then the corresponding figure

Do not include in results what is part of the methodology _lines 93-99; 108-109; 121-123; 143-144; 163; 181-183; 204-205; 238-239, 262-265; 288-290, 298-300; 307-310; 319-321.

Response: We have rearranged the order as suggested. We have moved these details to the "Materials and Methods" section as suggested.

Lines 246-248: Wouldn't this paragraph be more appropriate in discussion?

Response: This paragraph have been moved to discussion section as suggested.

Line 402: More important than comparing is explaining why this happened.

Response: We have attempted to provide an explanation in the subsequent statements.

Check some words are in upper case in the discussion.

Response: We have checked the words in the discussion as suggested.

At the end of the discussion, the future perspective could simply be included after the results obtained.

Response: We have added the “Future Perspective” section after “Conclusions” section.

Line 465:The description of the experiments with stresses was missing

Include the conclusion item

Response: We have described the experiments in Materials and Methods, and added the Conclusion item.

Round 2

Reviewer 1 Report

Comments and Suggestions for Authors

All problems have been replied to adequately.  However, the style of reference needs to be checked.

Author Response

All problems have been replied to adequately.  However, the style of reference needs to be checked.

Response: Thank you for your comment. We have now revised all references to adhere strictly to the journal's formatting guidelines.

Reviewer 2 Report

Comments and Suggestions for Authors

Dear Authors,

I appreciate the effort you have put into revising your manuscript, and I acknowledge the significant improvements made. However, I would like to bring your attention to a few remaining points that require further refinement:

  • Figure 2: Please add Tukey’s letters for the ethanol concentrations at 30%, 40%, and 50% to clarify statistical differences.

  • Figures 3, 4, 5, and 6: The quality of these figures needs improvement. I recommend increasing their resolution and adjusting their size or layout to enhance readability and clarity. Properly arranging figure components will help improve the overall presentation.

Addressing these points will further strengthen your manuscript. Thank you for your efforts, and I look forward to seeing the final version.

Author Response

I appreciate the effort you have put into revising your manuscript, and I acknowledge the significant improvements made. However, I would like to bring your attention to a few remaining points that require further refinement:

Figure 2: Please add Tukey’s letters for the ethanol concentrations at 30%, 40%, and 50% to clarify statistical differences.

Response: We have added Tukey’s letters for the ethanol concentrations at 30%, 40%, and 50% as suggested.

Figures 3, 4, 5, and 6: The quality of these figures needs improvement. I recommend increasing their resolution and adjusting their size or layout to enhance readability and clarity. Properly arranging figure components will help improve the overall presentation.

Response: We have improved the quality of Figure 3, 4, 5 and 6 as suggested.